# Gut Microbiota–Host Interactions in Inborn Errors of Immunity

**DOI:** 10.3390/ijms22031416

**Published:** 2021-01-31

**Authors:** Riccardo Castagnoli, Francesca Pala, Marita Bosticardo, Amelia Licari, Ottavia M. Delmonte, Anna Villa, Gian Luigi Marseglia, Luigi Daniele Notarangelo

**Affiliations:** 1Laboratory of Clinical Immunology and Microbiology, Division of Intramural Research, National Institute of Allergy and Infectious Diseases, National Institutes of Health, Bethesda, MD 20814, USA; riccardo.castagnoli@nih.gov (R.C.); francesca.pala@nih.gov (F.P.); marita.bosticardo@nih.gov (M.B.); ottavia.delmonte@nih.gov (O.M.D.); 2Department of Pediatrics, Foundation IRCCS Policlinico San Matteo, University of Pavia, 27100 Pavia, Italy; a.licari@smatteo.pv.it (A.L.); gl.marseglia@smatteo.pv.it (G.L.M.); 3Department of Clinical, Surgical, Diagnostic and Pediatric Sciences, University of Pavia, 27100 Pavia, Italy; 4Department of Molecular Medicine, University of Pavia, 27100 Pavia, Italy; 5San Raffaele Telethon Institute for Gene Therapy, Istituto di Ricovero e Cura a Carattere Scientifico (IRCCS) San Raffaele Scientific Institute, 20123 Milan, Italy; villa.anna@hsr.it; 6Istituto di Ricerca Genetica e Biomedica (IRGB), Consiglio Nazionale delle Ricerche (CNR), Milan Unit, 20133 Milan, Italy

**Keywords:** gut microbiota, inborn errors of immunity, primary immunodeficiencies, immune dysregulation, dysbiosis, gastrointestinal pathology

## Abstract

Inborn errors of immunity (IEI) are a group of disorders that are mostly caused by genetic mutations affecting immune host defense and immune regulation. Although IEI present with a wide spectrum of clinical features, in about one third of them various degrees of gastrointestinal (GI) involvement have been described and for some IEI the GI manifestations represent the main and peculiar clinical feature. The microbiome plays critical roles in the education and function of the host’s innate and adaptive immune system, and imbalances in microbiota-immunity interactions can contribute to intestinal pathogenesis. Microbial dysbiosis combined to the impairment of immunosurveillance and immune dysfunction in IEI, may favor mucosal permeability and lead to inflammation. Here we review how immune homeostasis between commensals and the host is established in the gut, and how these mechanisms can be disrupted in the context of primary immunodeficiencies. Additionally, we highlight key aspects of the first studies on gut microbiome in patients affected by IEI and discuss how gut microbiome could be harnessed as a therapeutic approach in these diseases.

## 1. Introduction

Inborn errors of immunity (IEI) are a group of disorders that are mostly caused by genetic mutations affecting immune host defense and immune regulation [1]. The clinical spectrum of IEI is extremely wide, with every organ being potentially involved. This clinical heterogeneity is due to the effect of different combinations of recurrent infections, autoimmunity, inflammatory manifestations, lymphoproliferation, atopy, and malignancy [1]. In this context, approximately one third of the 430 IEI reported in the most updated International Union of Immunological Societies (IUIS) classification presents various degrees of gastrointestinal (GI) involvement and for some IEI the GI manifestations represent the main and peculiar clinical feature [2,3]. Multiple etiologies contribute to GI pathology in IEI including infections related to the underlying host defense impairment, immune dysregulation, and malignancy [3]. Of note, intestinal homeostasis is determined by the complex interplay between the gut microbiota and the host immune system [4]. Gut microbiota plays a critical role in influencing the development and function of the host’s innate and adaptive immune system, which in turn shape microbiota composition and control maintenance of host-microbe symbiosis [5]. It is now known that, in genetically susceptible hosts, imbalances in microbiota-immunity interactions contribute to the pathogenesis of immune-mediated disorders [6,7]. In this context, the study of gut microbiota-host interactions in IEI may help understand the factors involved in the maintenance of intestinal homeostasis [8,9].

Here we focus on the interplay between gut microbiota and host immune system in IEI through a translational approach. We will also highlight some of the current knowledge regarding molecular mechanisms of gut microbiota–host interactions leading to intestinal homeostasis, as well as the disruption of these mechanisms in specific forms of IEI, as revealed by animal models and human diseases.

## 2. Overview on Molecular Mechanisms of Gut Microbiota–Host Interactions Leading to Intestinal Homeostasis

It is a well-known fact that in the human body there are as many microorganisms as human cells, so as to be acknowledged recently as a new organ: the human microbiome. This massive presence of foreign organisms, including bacteria, viruses and fungi, is possible thanks to the establishment of a fine balance between the microbiota and the immune system, which creates a favorable environment for both the host and the colonizing organisms. However, when altered by internal (i.e., genetic diseases, cancer) or external (i.e., antibiotics, diet, stress) factors, disruption of this balance may result in pathology. Gut microbiota–host interactions are fundamental for the education and proper function of the immune system (reviewed in [10]). This homeostatic interaction is reached through a continuous dialogue of positive and negative feedbacks between the gut microbiota and the various modules of the innate and adaptive immune cells, here briefly reviewed and summarized in Figure 1.

### 2.1. Epithelial Interaction and Compartmentalization of Gut Microbiota

Minimal contact between microorganisms and the epithelial surface is fundamental to establish a homeostatic relationship in the gut and avoid massive microbial translocation. A combination of mucus, antimicrobial peptides, IgA secretion, epithelial and immune cells in the intestinal mucosa all contribute to protect from microbial invasion of the body milieu. Intestinal mucus represents a physical barrier between the microbes and the mucosa, thereby limiting their interactions and the chances of a microbial translocation [11]. Antimicrobial peptides (AMPs) are one of the phylogenetically oldest systems of innate immunity and are mainly produced by Paneth cells [12]. These proteins can have direct functions, including disruption of the bacterial inner membrane or enzymatic activity against the bacterial cell wall, and can be constitutively expressed by epithelial cells [13]. Their production can also be induced by stimulation of pattern-recognition receptors (PRRs) by commensal-derived products, as in the case of RegIIIγ, a lectin produced in a MyD88-dependent manner and with a direct microbicidal effect on gram-positive bacteria [14]. The control of the intestinal mucosa is thus exerted by the epithelial cells, which are able to integrate and respond to various stimuli coming through their PPRs, such as Toll-like receptors (TLRs) [15], Nod-like receptors (NLRs) [16] and short-chain fatty-acid receptors (SCFARs) [17], to establish the so-called “demilitarized zone’’ [18] (Figure 1A).

### 2.2. Gut Microbiota–Innate Immunity Homeostatic Interplay

Innate immunity represents the first line of defense against foreign threats, as it can quickly recognize and respond to pathogens. This line of defense includes monocytes and macrophages, as well as non-classical lymphocytes like innate lymphoid cells (ILCs), γδT cells, mucosal-associated invariant T (MAIT) cells, invariant natural killer T (iNKT) and NK cells (Figure 1A). It has been shown that monocytes and macrophages can be influenced and instructed by microbiome-derived metabolites. Short-chain fatty acids (SCFA) like butyrate can induce gene expression changes and imprint an antimicrobial program in macrophages [19,20], and can drive monocyte-to-macrophage differentiation via histone deacetylase (HDAC) inhibition [21]. Similarly, trimethylamine N-oxide (TMAO) can induce M1 macrophage polarization in a NLR-dependent manner [22]. ILCs are enriched in mucosal tissues and are specialized in cytokine and chemokine production against infectious agents, as well as in promoting tissue repair [23]. An example comes from ILC3s, which produce IL-22 in response to *Helicobacter* species and segmented filamentous bacteria (SFB). IL-22 plays a pivotal role in promoting AMP and mucus production, while enhancing epithelial regeneration and wound repair [24,25,26,27]. Additionally, MHCII-expressing ILC3s can interact with and deplete CD4+ T cells specific for intestinal commensal bacteria [28,29]. γδT cells are abundant at barrier sites and are believed to have a prominent role in the recognition of lipid antigens, which are often microbiota-derived metabolites [30]. γδT cells have been shown to be able to expand in an IL-1- and/or IL-23-dependent manner thanks to commensal colonization in these sites [31,32]. MAIT cells express a semi-invariant T-cell receptor restricted by a single non-polymorphic and highly evolutionarily conserved MHC class I-related protein 1 (MR1) [33]. They are dependent on the microbiota, as they are absent in germ-free (GF) mice, and recognize intermediates of the microbial riboflavin synthesis pathway [33,34]. Once activated by these metabolites, MAIT cells acquire cytotoxic capacity in a perforin- and granzyme/granulysin-dependent manner, as well as producing inflammatory cytokines such as IFNγ, TNFα, GM-CSF, and IL-17 [35,36]. Lastly, gut microbiota has been shown to interact with iNKT. Studies performed in GF mice have shown that exposure early in life to sphingolipids originating from commensals like *Bacteroides fragilis* restores iNKT numbers and is protective against colitis [37,38].

### 2.3. Gut Microbiota–Adaptive Immunity Homeostatic Interplay

In recent years, several studies have shed light on the mechanisms of homeostatic interaction and mutualism between the microbiome and the adaptive immune system (Figure 1B). B cells play a major role in controlling the intestinal mucosa by producing a large pool of secretory IgA antibodies able to recognize commensal bacteria [39]. Secretory IgA are produced through both T cell-independent and -dependent mechanisms [40]. T-independent production of IgA can exclude microorganisms from the gut, while adaptive IgA responses play a major role in shaping the microbial community itself and its mutualistic relationship with the immune system [41,42]. The diversification of the IgA repertoire allows for diversification among the different microbial species in the gut, driving the expansion of Foxp3+ regulatory T cells (Tregs), which will, in turn, sustain IgA production in the germinal centers creating a feed-forward loop [41]. Of note, Tregs can be peripherally induced in the gut by oral antigens and have been shown to have a distinct repertoire from natural Treg (nTreg) cells generated in the thymus [43,44]. IgA diversification happens in the Peyer’s patches, where intestinal dendritic cells (DCs) present commensal-derived antigens associated with the epithelium to both B and T cells [45]. The microenvironment of Peyer’s patches germinal centers, rich in cytokines, chemokines and various metabolites, drives B cells to proliferate and undergo class-switch to produce preferentially IgA from IgM [46,47]. In addition to the Peyer’s patches, DCs loaded with microbial antigens travel to the mesenteric lymph nodes to instruct more B cells, so that the IgA response is kept compartmentalized. The secreted IgAs are transcytosed across epithelial cells [42], where they contribute to impede bacteria attachment to the mucosa. Different T cell subsets reside in the lamina propria of the gut in a mutualistic relationship with the microbiome. For example, Th17 differentiation and cell frequencies are severely reduced in GF mice [48]. Interestingly, Th17 cells can assume pro-inflammatory or anti-inflammatory properties based on the class of bacteria they interact with. The first case is driven by *Citrobacter*, while SFB elicit a non-inflammatory Th17 response [49]. Similarly, a recent study has shown that activated CD8+ (cytotoxic) T cells depend on microbiota-derived SCFAs to transition to memory cells, as this differentiation is absent in GF mice [50]. We have already mentioned that SFB induce specific responses in several lymphoid cells. Interstingly, SFB colonization is age-dependent in humans, with most individuals colonized within the first 2 years of life, but this colonization disappears by the age of 3 years [51]. Another example comes from RORγt-expressing CD4+ T cells which accumulate within the mucosa upon SFB colonization. These cells produce IL-17A in response to epithelial-derived serum amyloid A proteins 1and 2 (SAA1/2) [27,52]. Serum amyloid A also works as a carrier of both high-density lipoprotein and retinol, delivering these molecules to APCs and amplifying the induction of IL-17 (Th17) responses within tissues [53]. Follicular helper T (Tfh) cells are essential for the germinal center formation, affinity maturation, class-switch, and memory development of B cells [54]. In the intestinal mucosa, they play a pivotal role in maintaining homeostasis. The absence of the co-receptor PD-1 on these cells can alter the microbiota composition [55]. At the same time, microbiota can induce Tfh differentiation. In particular, SFBs can promote Tfh differentiation by limiting access of CD4+ cells to IL-2, thereby enhancing the expression of the Tfh master regulator Bcl-6 [56]. Tfh differentiation depends also on MyD88 signaling; accordingly, in the absence of microbial stimuli, it is inhibited in GF settings [57]. These (and other) studies point towards the fundamental role of microbiota in the induction, education, and function of the host immune system.

## 3. Gut Microbiota–Host Interactions in Animal Models of Inborn Errors of Immunity

Most of our knowledge on the establishment of homeostatic immunity at the intestinal mucosa derives from the study of animal models, in which systematic manipulation of the microbiota (GF and single microbe/metabolite inoculations) and of the immune system (including animal models of IEI) has allowed the dissection of the mechanisms involved in the disruption of gut microbiota-host interactions. At the same time, the characterization of these mechanisms may prompt the development of novel therapeutic approaches in patients. However, it is fundamental to consider that most inbred laboratory animals have limited exposure to microorganisms and little diversity in their microbiota compared to wild ones. A recent study by Rosshart et al. showed that “wildlings” mice obtained by implanting lab-strain embryos into wild mice had a systemic immune phenotype and a bacterial, viral and fungal microbiome much closer to those of their wild counterparts [58].

In the context of inborn errors of immunity, the impairment of immunosurveillance as well as immune dysfunction and autoimmunity in the gut, may favor dysbiosis, leading to mucosal permeability and inflammation. While the systemic consequences of immune defects in animal models of IEI have been extensively studied [59], the specific effects in the gut and in particular on the gut microbiota are still ill-defined. A recent study by Zheng et al. used SCID mice and NOD/SCID mice (with defects restricted to adaptive immunity, or involving both adaptive and innate immunity, respectively) to investigate the interactions between the host immune system and the gut microbiome [60]. Their metagenomic analysis revealed a decrease in bacterial diversity in both models, defined by alpha diversity. Alpha diversity is a measure of the diversity within a microbial community (in opposition to beta diversity which measures differences between communities) and takes into account the number of observed operational taxonomic units (OTUs) (richness) and the relative abundances of OTUs (evenness). This decrease was associated with the development of inflammatory complications and infections as a consequence of microbial dysbiosis. Immunodeficient mice presented with a significant increase in SCFA-producing bacteria, accompanied by an overall decrease in the concentration of secreted immunoglobulins, mostly IgA. The lack of T and B cells in the Peyer’s patches of these mice, as well as the limited availability of secretory IgA, could lead to increased microbial translocation [61]. In particular, the role of secretory IgA in limiting the contact of commensals to the epithelium has been highlighted in studies performed in mice lacking IgA [41,46,55]. These mice presented with lower microbial alpha diversity and were unable to control SFB attachment to the gut epithelium, leading to immune activation. A similar phenotype was reported in Activation-induced cytidine deaminase-deficient (*Aicda^−/−^*) mice, in which impaired class switch recombination (and therefore lack of IgA) is associated with the expansion of SFB and *Clostridium* spp. in their small intestine [46,62]. Correlation between gut dysbiosis and gut inflammation has also been shown in other models of IEI, such as RAG deficiency, IPEX, and IL-10-deficiency [63,64,65]. The recombination-activating gene 1 (RAG1) and RAG2 proteins initiate V(D)J recombination; null mutations in these genes result in the absence of B and T cells. However, a wide array of *RAG* mutations has been described in humans that can lead to different levels of residual recombination activity [66,67,68]. These hypomorphic mutations cause different degrees of immunodeficiency associated with multisystem autoimmune–like manifestations mediated by oligoclonal self-reactive T and B cells. The effects of residual presence of B and T cells on the intestinal mucosa and the microbiome has been investigated in a mouse model of hypomorphic RAG2 deficiency (*Rag2^R229Q/R229Q^* mice) by Rigoni et al. [63]. These mice manifested spontaneous gut inflammation and epithelial disruption characterized by dominant mixed Th1/Th17 immune responses, together with a local B cell deficiency. Bacterial diversity was significantly reduced when compared to wild-type mice, with a preferential enrichment for *Proteobacteria*. High levels of endotoxin in the serum indicated a defect in the permeability of the gut-blood barrier, likely due to B cell and IgA deficiency. Excessive inflammation in the gut could be partially rescued by administration of broad range antibiotics [63]. The *scurfy* (*sf*) mouse is used as a model of immune dysregulation, polyendocrinopathy, enteropathy, X-linked (IPEX) syndrome, due to mutations in the FOXP3 gene. The gene encodes an essential transcriptional regulator that plays a pivotal role in the development and function of regulatory T cells (Tregs) [69,70]. These mice present with autoimmunity due to the absence of functional Tregs and develop severe inflammation leading to early death [71]. A recent study showed that antibiotics impact positively on the survival and lethal inflammation in *sf* mice despite their effect of reducing microbial diversity [64]. Amelioration of inflammation correlated with depletion of IL-6, which may be regulated by gut microbiota. Additionally, He et al. proposed that accurate characterization of microbiota and microbial-derived metabolites could be employed to develop novel therapeutic strategies for patients with immunoregulatory diseases such as IPEX syndrome [64]. A mutualistic relationship between microbiota and immune cells has been shown in the *Il10^−/−^* mice. When born and maintained in GF conditions, *Il10^−/−^* mice do not develop the spontaneous enterocolitis typical of the model [72], and similar effects have been observed when *Il10^−/−^* mice are treated with antibiotics since neonatal age [73]. Additionally, different species had different effects on the immune system. For example, members of the *Lactobacillus* and *Bifidobacterium* species attenuated gut inflammation in *Il10^−/−^* mice, whereas *E. faecalis* and *Helicobacter* worsened their phenotype [74,75,76]. Drawing parallels between human and mouse model systems has served as a foundation to define common pathways utilized by microbes and the host during health and disease. The power of these models relies on the possibility to manipulate single genes as well as single microbial species [77]. The rapid development of microbiome sequencing technologies, like 16S rRNA sequencing and shotgun metagenomic, will further expand our knowledge of gut microbiota–immunity interactions and harness our ability to treat mucosal diseases.

## 4. Gut Microbiota–Host Interactions in Human Inborn Errors of Immunity

Monogenic diseases affecting the immune system represent unique models to dissect gene functions and biological pathways, and may therefore provide important insights into the mechanisms of how the immune system works *in vivo*. Several IEI affect the immunological pathways involved in the above-described mechanisms of gut microbiota–host interactions, disrupting intestinal homeostasis. Interestingly, while microbiota perturbations in polygenic inflammatory bowel disease (IBD) can have a complex etiology, dysbiosis in patients with IEI is driven primarily by the gene defects.

In order to retrieve all publications analyzing gut microbiota-host interactions in human IEI, the literature review has been performed employing EMBASE, Pubmed, Scopus and Web of Science databases. The search strategy was performed using a free-text search (keywords: inborn errors of immunity, primary immunodeficiency, microbiota, microbiome, dysbiosis, gut, intestinal homeostasis) and thesaurus descriptors search (MeSH and Emtree), adapted for all the selected databases. We searched all articles published up to December 2020. The inclusion criteria for eligible articles were the following: publication in peer-reviewed journals and the English language. Articles were excluded by title, abstract, or full text for irrelevance to the analyzed topic. Lastly, to identify further studies that met the inclusion criteria, the references of the selected articles were also reviewed. Table 1 shows a summary of the available evidence.

### 4.1. Immunodeficiencies Affecting Cellular and Humoral Immunity

Severe combined immune deficiency (SCID) and combined immunodeficiency (CID) represent a heterogeneous group of genetic disorders characterized by impairment of T cell development and/or function [78,79,80]. In some forms of SCID, the number of circulating B and/or NK cells is variably affected, but B cell function is consistently impaired because of poor helper T cell activity. SCID usually present in the first year of life with severe opportunistic infections, failure to thrive and chronic diarrhea [78]. In the absence of therapy (specifically, hematopoietic stem cell transplantation (HSCT) [81], gene therapy (GT) [82] or enzyme replacement therapy [83]), patients with SCID usually die within the first years of life. The pre-symptomatic identification of infants with SCID through newborn screening has significantly improved the clinical management of these conditions [84]. CID is also characterized by increased susceptibility to recurrent and life-threatening infections; however, because of residual T cell function, clinical onset is often delayed (>1 year of age) as compared to SCID, and disease severity may also be decreased [85,86,87]. Furthermore, at variance with SCID, CID is also characterized by an increased risk of immune dysregulation and malignancy. Of note, some forms of SCID and CID include extra-hematopoietic manifestations reflecting broader tissue distribution of the gene product.

To better understand the role of gut microbiota–host interactions in SCID patients, Lane et al. examined changes in the gut microbiota in three patients with SCID (2 X-SCID and 1 RAG deficiency) treated by HSCT [88]. This pilot study was the first to analyze the gut microbiota in SCID and demonstrated differences in gut microbial composition before and after HSCT. The small number of patients does not allow to draw firm conclusions; however, low microbial diversity associated with dominance of some species (mainly *Escherichia*, *Enterococcus* and *Staphylococcus*) in the course of HSCT was apparent [88]. Along with changes in microbial abundance, qualitative variations in microbial products, i.e., metabolites, have been described. In particular, metabolomic analysis revealed intra-individual differences between pre- and post-HSCT samples in SCID patients [89]. Although the possible immunoregulatory role of these metabolites is still under investigation, gut microbial/metabolic signatures associated with beneficial or pathogenic potential may represent novel biomarkers for monitoring intestinal and systemic inflammation during HSCT. Of note, the effect of gut microbiota in HSCT is an active research field, also in diseases other than IEI [90,91]. Compelling evidence shows that the gut microbiota has a role in the development of graft-versus-host disease (GvHD) both in adult and pediatric patients [92,93], but the exact mechanisms of this relationship need to be elucidated [94]. In this context, the manipulation of the gut microbiota, through pharmacologic modification/decontamination [95,96] or fecal microbiota transplantation (FMT) [97,98], may shape the microbiota composition and modulate immune responses after HSCT to benefit the host. Changes in gut microbiota in patients treated by GT are another area of active investigation. Recently, Clarke et al. analyzed the parallel development of the microbiome and of the immune system in six patients with X-SCID treated by GT [99]. Interestingly, in this cohort, the gut microbiota of immune reconstituted patients resembled that of healthy children and these changes were concomitant with the progressive normalization of the T-cell receptor (TCR) repertoire. 

Among CIDs, a recent study by Zhang et al. reported on fecal microbial dysbiosis in children with Wiskott-Aldrich syndrome (WAS) [100]. WAS is an X-linked IEI characterized by immunodeficiency, eczema, and thrombocytopenia with small platelets [101]. WAS is caused by mutations in the *WAS* gene, encoding the WAS protein (WASP), which is involved in signal transduction and cytoskeleton remodeling [102]. A broad spectrum of clinical phenotypes has been described in patients with *WAS* mutations, including recurrent and/or chronic infections, autoimmune manifestations, and increased susceptibility to malignancies, especially EBV-associated lymphoma. Moreover, up to 5–10% of these patients develop IBD which may present as Crohn’s disease or ulcerative colitis [103]. Zhang et al. showed reduced fecal microbial community richness and diversity in WAS patients compared to age-matched healthy controls [100]. The relative abundance of *Bacteroidetes* and *Verrucomicrobia* in WAS was significantly lower, while that of *Proteobacteria* was markedly higher. Among WAS children, those with IBD and those who failed to express WASP, presented with more severe microbial dysbiosis. Interestingly, abnormalities of fecal microbiota were similar to those observed in polygenic IBD [104,105,106,107,108,109,110,111], suggesting that WASP may play crucial function in microbial homeostasis and that microbial dysbiosis may contribute to IBD in WAS. Moreover, these microbial alterations may represent novel targets for monitoring and managing intestinal inflammation in WAS.

### 4.2. Predominantly Antibody Deficiencies

Common variable immunodeficiency (CVID) is a clinically heterogeneous primary immunodeficiency characterized by hypogammaglobulinemia and impaired antibody responses to vaccine antigens [112,113]. CVID typically manifests with recurrent respiratory tract infections [114]; gastrointestinal involvement is common and may mark clinical onset of the disease [115,116]. In addition to immunodeficiency, more than half of all CVID patients develop non-infectious complications such as autoimmune diseases, lymphoproliferation, granulomatous disease and malignancies, with significant morbidity and mortality [117,118]. A genetic cause has been identified in 10–30% of CVID patients, depending on the cohort analyzed [117,119,120], including mutations in *NFKB1, NFKB2, ICOS, TNFRSF13B, TNFRSF13C, CD19, CR2, MS4A1, CD81, IL21, LRBA, CTLA4, PRKCD*, and *IKZF1*. Although these cases provide genetic insights into disease pathogenesis, the role of environmental factors is possibly critical and still poorly defined. GI complications are reported in 20–60% of CVID patients, with infectious diarrhea occurring in 30–50% of cases [3]. Other GI manifestations include inflammation of the small or large intestine resembling ulcerative colitis or Crohn’s disease, villous flattening resembling celiac disease, collagenous enterocolitis, pernicious anemia, nodular lymphoid hyperplasia, lymphoma or gastric adenocarcinoma [121,122,123]. Non-infectious complications of CVID are mainly caused by immune dysregulation. This chronic and aberrant systemic immune activation seems to be, at least in part, a consequence of increased microbial translocation [61,124,125]. Jørgensen et al. investigated gut microbiota composition in 44 CVID patients, 45 patients with IBD and 263 healthy controls [126]. Microbial alpha diversity has been used to evaluate differences among the three groups. High alpha diversity with a large number of microbial species of roughly even abundances was documented in healthy subjects. By contrast, microbiota of CVID patients was characterized by reduced alpha diversity, correlating with increased levels of plasma lipopolysaccharide (LPS), a marker of microbial translocation, increased levels of soluble (s) CD14 and sCD25 (biomarkers of immune activation), and reduced IgA serum levels. Importantly, these abnormalities were only observed in CVID patients with immune dysregulation, as LPS levels and alpha diversity were comparable between healthy controls and CVID patients without immune dysregulation. Moreover, microbiota alpha diversity was comparable in CVID patients and in those with IBD, but differences in taxonomic profile between the two diseases were reported. Also, differently from what observed in CVID, alpha diversity did not correlate with markers of endotoxemia in IBD patients. This may suggest that specific changes in microbiota composition, rather than broad differences in diversity, are responsible for increased microbial translocation in CVID [61]. In another study, mucosal IgA deficiency in CVID has been associated with GI inflammation and malabsorption [127], providing a link between IgA deficiency in CVID and mucosal barrier damage that may facilitate microbial translocation [128]. However, microbial translocation was not reported in patients with selective IgA deficiency (sIgAD), with one study reporting no increase in serum LPS levels [129,130]. Considering these results, it has been proposed that in sIgAD systemic microbial translocation (evaluated as serum LPS level) may be prevented by compensation through IgG or IgM. In this context, Fadlallah et al. showed that IgM can compensate sIgAD for some, but not all bacteria. IgM are efficient in binding *Actinobacteria* (including *Bifidobacterium* spp.), but only a limited range of *Proteobacteria* (excluding *Enterobacteriaceae*) and *Firmicutes* (including *Clostridium* and *Faecalibacterium* spp.) [129]. However, adequate IgM and/or IgG levels in sIgAD seem to protect from endotoxemia, while this compensatory effect is lacking in CVID. In addition, a recent study by Fadlallah et al. [131] demonstrated that IgG from healthy individuals recognize the microbiota of CVID patients much less effectively than the microbiota of healthy subjects, suggesting that IgG supplementation may be insufficient to limit microbiota dysregulation and/or translocation in CVID.

Regarding the role of gut metabolites in inducing inflammation in CVID patients, Macpherson et al. recently showed that high concentrations of the gut microbiota-dependent metabolite trimethylamine N-oxide (TMAO) is associated with systemic inflammation and increased gut microbial abundance of *Gammaproteobacteria* in CVID patients, suggesting that TMAO could represent a link between systemic inflammation and gut microbial dysbiosis. In this setting, gut microbiota composition may be targeted to reduce systemic inflammation in CVID [132]. 

Finally, several studies are under way to investigate the effect of antibiotic treatment on gut microbiota composition and eventually on systemic inflammation in CVID. Although use of antibiotics in the year preceding microbiota sampling was not found to alter the microbiota in one CVID study [126], cumulative effects of long-term antibiotic use, as is often the case in IEI, can not be excluded. Recently, Jørgensen et al. [133] evaluated the effect of a single broad-spectrum antibiotic (rifaximin) with the aim to correct microbial dysbiosis in CVID. No effects were recorded on microbial translocation (serum LPS), immune cell activation (sCD15 or sCD25), and immune dysregulation. These results suggest that antibiotic monotherapy is not a promising strategy to modulate gut microbiota in CVID patients.

### 4.3. Studies in Other Human Inborn Errors of Immunity

Few other studies evaluated the gut microbiota-host interactions in other human IEI. Fecal microbiota and association with IBD were analyzed in patients with chronic granulomatous disease (CGD, 10 patients), X-linked inhibitor of apoptosis (XIAP) deficiency (six patients) and partial tetratricopeptide repeat domain 7A (TTC7A) deficiency (five patients), and compared to non–genetically-determined/polygenic IBD (18 patients) and 23 healthy subjects [134]. CGD is caused by gene mutations that affect the function of the nicotinamide adenine dinucleotide phosphate (NADPH) complex, resulting in defective production of microbicidal reactive oxygen species (ROS) [135,136]. The most frequent form of CGD is inherited as an X-linked trait (XL-CGD) and is due to mutations of the *CYBB* gene encoding for the gp91^phox^ subunit of the NADPH oxidase complex. Autosomal recessive (AR) forms (AR-CGD) are caused by mutations of the genes that encode for the p22^phox^, p47^phox^, p67^phox^, and p40^phox^ subunits. The main clinical features of CGD encompass recurrent bacterial and fungal infections and a high rate of inflammatory complications [137,138]. Gastrointestinal involvement with features common to both Crohn’s disease and ulcerative colitis is a common initial feature of CGD and precedes the diagnosis in up to 17% of the patients [139,140,141,142]. The reported prevalence of CGD-associated IBD (CGD-IBD) ranges from 31% [143] to 88% [144], being one of the main clinical manifestations especially in patients with p40^phox^ deficiency [145]. A recent report from the Primary Immune Deficiency Treatment Consortium (PIDTC) showed that HSCT is curative for CGD colitis, and importantly, that colitis itself is not a contraindication to HSCT as it does not lead to an increased risk of mortality [146].

XIAP deficiency, also known as X-linked lymphoproliferative disease type 2 (XLP2), is due to mutations of the *XIAP* gene, that controls cell survival and inflammatory responses [147,148]. Patients with XIAP deficiency are at high risk for hemophagocytic lymphohistiocytosis [149] and usually present with inflammatory manifestations, including Crohn’s-like bowel disease [150]. Other variable clinical manifestations include severe infectious mononucleosis, splenomegaly, fistulating skin abscesses, and antibody deficiency contributing to recurrent infections [147].

Mutations of the *TTC7A* gene have been identified in patients with multiple intestinal atresias and other related disorders, including CID and very early onset IBD with apoptotic enterocolitis [151,152,153]. Pathological findings include structural abnormalities of the enterocytes, increased apoptosis of intestinal cells, reduced proliferation of intestinal crypts, and defects of thymic architecture associated with lymphoid depletion [154,155].

Sokol et al. showed that the gut microbiota of patients with CGD, XLP2 and TTC7A deficiency has distinct alterations, supporting the notion that defects in these genes perturbing immune function may favor intestinal dysbiosis [134]. Of note, significant differences of microbiota were detected when comparing patients with the same gene defect who differed for the presence or absence of GI involvement. Patients with TTC7A deficiency exhibited an increase in *Proteobacteria*, including *Gammaproteobacteria* and *Epsilonproteobacteria*. However, bacteria from the *Ruminococcaceae* family and notably the *Oscillospira* genera were decreased. Patients with XIAP deficiency showed an increased abundance of *Proteobacteria*, *Firmicutes*, *Actinobacteria*, and *Fusobacteria* phyla. Four of these taxa (*Fusobacterium*, *Scardovia*, *Veillonella* and *Rothia dentocariosa*) are known members of the oral microbiota, are not usually found in the gut and have been reported in IBD, colorectal cancer, and liver diseases [156]. Patients with CGD exhibited an increased abundance of *Ruminococcus gnavus*, which has been associated with ileal Crohn’s disease [157]. Overall, this study has been the first to evaluate gut microbiota in IEI with IBD.

Recently, Xue et al. identified intestinal dysbiosis in pediatric Crohn’s disease patients with Interleukin 10 Receptor Subunit Alpha (*IL10RA*) mutations [158]. Patients with IL-10 receptor or IL-10 deficiency usually present in the first few months of life with severe IBD, often characterized by perianal disease and enterocutaneous and recto-vaginal fistulas [159,160]. Microbiota composition was investigated in 17 patients with *IL10RA* mutations, 17 patients with pediatric Crohn’s disease, and 26 healthy children [158]. Both patients with IL-10RA deficiency and those with Crohn’s disease showed a decrease in diversity of gut microbiome. The relative abundance of *Firmicutes* was substantially increased in the *IL10RA* group. Interestingly, in patients with *IL10RA* mutations, the degree of gut dysbiosis (calculated based on the relative abundance of five taxa at the order level: *Lactobacillales, Micrococcales, Veillonellaceae, Clostridiales, and Selenomonadales* [161]) appeared to be directly associated with disease severity.

Finally, a recent report showed the beneficial effect of fecal microbiota transplantation (FMT) before HSCT in a child with IPEX [162]. IPEX is characterized by multisystem autoimmunity, including severe enteropathy, autoimmune cytopenia, type I diabetes mellitus, hypothyroidism, and other autoimmune symptoms, in which the earliest and most prominent manifestation is chronic diarrhea [163,164,165]. Wu et al. reported marked decreased diversity in gut microbiota composition in the child with IPEX compared to healthy controls. After receiving FMT treatment, a remission of the diarrhea without apparent side effects was observed. Stool output was significantly reduced, concurrent with increased microbial diversity and modification of his microbiota composition. The patient finally achieved complete recovery after HSCT [162]. This report suggests an association between the gut microbiota and clinical symptoms of IPEX and, for the first time, proposed FMT as an alternative therapy for severe diarrhea unresponsive to routine therapy in these patients. Of note, most experience with FMT has been in the treatment of recurrent *Clostridium difficile* infections; patients with IEI were included in four of these studies [166,167,168,169] two of which reported lack of safety concerns and successful resolution of *C. difficile* [168,169]. However, further studies are needed to evaluate the efficacy and safety of FMT in IEI.

## 5. Conclusions

Host immunity and microbiota interplay is a crucial symbiotic and dynamic relationship. Several lines of investigation have highlighted how the immune system plays a central role in shaping the composition of the microbiota. At the same time, resident commensals provide signals that induce normal immune system development and instruct the ensuing immune responses. As we continue to shed light on the cellular and molecular mechanisms of currently known IEI and describe new ones, additional insights into the interactions between host-immunity, the microbiome, and gut function are emerging. The advent of -omic technologies, including shotgun metagenomics, metatranscriptomics, and metabolomics, has created a high volume of complex data to more deeply characterize phenotypes of both the microbiome and the host, and in turn unveil potential pathobiological mechanisms as well as therapeutic strategies. Although only a few studies of the microbiome in patients with IEI have been performed to date, the interest in this field is rapidly growing. By studying IEI, we can understand which cells and molecular mechanisms are fundamental for immune function at barrier sites as well as non-redundant pathways required for intestinal colonization by commensal microbes. One common characteristic of the IEI microbiome is the decrease in microbial species typically associated with health. Intestinal commensals directly antagonize the proliferation of pathogenic bacteria and favor immune mechanisms to suppress competing microbes, overall activating responses that maintain epithelial barrier integrity. Alterations of the gut microbiota-host interactions may underlie aberrant immune responses in IEI. However, independent and ideally larger and longitudinal studies are required to confirm these findings. Considering the potential confounding factors such as sex, age, diet, treatment, geographical location, socioeconomic features, and gastrointestinal symptoms is particularly difficult, yet fundamental, in studies aiming at discriminating genetic factors from environmental influences. Of note, IEI patients frequently require antimicrobial therapy to manage or prevent chronic or recurrent infection. The specific effect of long-term antibiotic use on the diversity of the human microbiome is still not sufficiently understood, especially not in relation to immune dysregulation. Finally, given the reported relationship between gut microbiota composition and IEI, the use of therapeutic intervention to correct intestinal dysbiosis may hold promise. The manipulation of the gut microbiota, through pharmacologic modification/decontamination or FMT, may shape the microbiota composition, depleting pathogenic bacteria and/or reconstituting missing health microbes, overall favoring intestinal homeostasis. However, to benefit for the gut microbiota as a target and an instrument of therapy, it is fundamental to fully understand the gut microbiota-host interactions in patients with IEI and clearly demonstrate the efficacy and safety of these procedures.

## Figures and Tables

**Figure 1 ijms-22-01416-f001:**
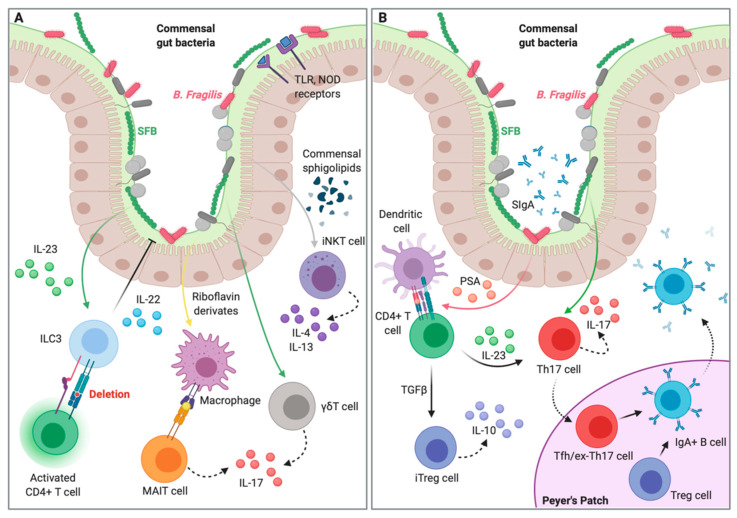
Key host–microbiota interactions in homeostasis. (**A**) Microbiome-derived metabolites and TLR/NOD ligands can be picked up directly by epithelial cells and immune cells. Riboflavin metabolites are presented by macrophages to MAIT cells, promoting their maturation and cytotoxic activity. Commensal-induced cytokines promote IL-17A production from γδT cells. ILC3-derived IL-22 prevents bacteria translocation to the lamina propria. Moreover, MHCII-expressing ILC3 deplete commensal-specific CD4+ T cells to prevent immune responses against harmless commensals. Early life colonization limits iNKT expansion within the colonic lamina propria and lungs via commensal sphingolipids. (**B**) Intestinal colonization by segmented filamentous bacteria (SFB) promotes Th17 cell differentiation. SFB colonization drives an ILC3/IL-22/SAA1/2 axis that licenses IL-17A production. Foxp3+ Treg cells and Tfh/ex-Th17 cells localize in the Peyer’s patches and promote B cell class-switch and production of IgA, which are then secreted in the intestinal mucosa. Colonization with *Bacteroides fragilis* promotes CD4+ T cell differentiation via polysaccharide A (PSA) presentation from dendritic cells. In the presence of TGFβ CD4+ T cells can differentiate in induced Treg (iTreg) and produce IL-10 to promote tissue homeostasis. Created with BioRender.com.

**Table 1 ijms-22-01416-t001:** Evidence of the role of gut microbiota–host interactions in human inborn errors of immunity.

Disease	Genetic Defect	Inheritance	Main Findings	Reference
Immunodeficiencies Affecting Cellular and Humoral Immunity (Including CID with Associated or Syndromic Features)				
SCID	*IL2RG* (X-SCID) *RAG1*	XL AR	Gut microbiota and fecal metabolite composition can be differentiated into pre- and post-HSCT groups. Gut microbiota of X-SCID patients changes to more resemble those of healthy children after successful gene therapy.	[88,89] [99]
Wiskott-Aldrich syndrome	*WAS*	XL	Reduced fecal microbial community richness and diversity in WAS patients compared to age-matched healthy controls. Among WAS children, those with IBD and those who failed to express WASP, presented with more severe microbial dysbiosis.	[100]
Predominantly antibody deficiencies				
CVID	Multiple genetic defects	Variable	In CVID patients with immune dysregulation, reduced microbial alpha diversity correlates with increased levels of LPS, soluble CD14 and CD25, and reduced IgA serum levels. IgG from healthy subjects targets the microbiota of CVID patients much less effectively than the microbiota of healthy subjects. Elevated concentrations of the gut microbiota-dependent metabolite TMAO is associated with systemic inflammation and increased gut microbial abundance of *Gammaproteobacteria* in CVID patients A single broad-spectrum antibiotic (rifaximin) does not modify microbial translocation, immune cell activation, and immune dysregulation	[126] [131] [132] [133]
sIgAD	Unknown	Unknown	Adequate IgM and/or IgG induction in sIgAD may protect from endotoxemia, while this compensatory response is lacking in CVID.	[129,130]
Other IEI				
CGD XIAP deficiency TTC7A deficiency	*CYBB, * *CYBA, CYBC1,* NCF1, NCF2, NCF4 *XIAP* *TTC7A*	XL AR XL AR	Gut microbiota of patients with different genetic defects has distinct alterations; moreover, patients with the same gene defect who differ for the presence or absence of GI involvement display different microbial communities.	[134]
IL-10 receptor deficiency	*IL10RA*	AR	The degree of gut dysbiosis (calculated based on the relative abundance of five taxa at the order level: *Lactobacillales, Micrococcales, Veillonellaceae, Clostridiales, and Selenomonadales*) appears to be directly associated to disease severity.	[158]
IPEX	*FOXP3*	XL	First report on the effect of FMT before HSCT in a child with IPEX	[162]

AR, autosomal recessive; AD, autosomal dominant; CGD, chronic granulomatous disease; CVID, common variable immunodeficiency; FMT, fecal microbiota transplantation; HSTC, hematopoietic stem cell transplantation; IEI, inborn errors of immunity; IL-10, interleukin-10; IPEX, immune dysregulation, polyendocrinopathy, enteropathy, X-linked; LPS, lipopolysaccharide; sIgAD, selective IgA deficiency; SCID, severe combined immunodeficiency; TMAO, trimethylamine N-oxide. TTC7A, Tetratricopeptide Repeat Domain 7A; XIAP, X-linked inhibitor of apoptosis; XL, X-linked.

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
