# Peer review of "Gut Microbiota–Host Interactions in Inborn Errors of Immunity"

_ijms, 2021, doi:10.3390/ijms22031416_

Round 1
Reviewer 1 Report
The review written by Castagnoli et al. provides a broad and well written overview of 'Gut Microbiota-Host Interactions in IEI'. It provides an interesting and timely perspective on this particular aspect of microbiome-host immunity dialogue.
Minor points:
- lines 125-127: half the sentence seems to be missing. The authors probably wished to say: "...express a semi-invariant T cell receptor restricted by a single non-polymorphic ..." or something along these lines.
- Also related to MAIT cells, the authors need to slightly modify Figure 1. The riboflavin metabolites driving MAIT cell activation are chemically unstable and are unlikely to be present in free form in the intestinal lumen. I would suggest to remove the yellow round symbols; keep the yellow arrow (which can originate from a bacteria) and add the text "riboflavin derivates" next to the arrow, but on the lamina propria side of the arrow, without any round symbols.
- lines 132-135: the authors seems to amalgamate iNKT and NK cells. Readers not well versed in immunology may believe that the interaction between microbial lipids and iNKT cells, as discussed in the second sentence, may also apply to NK cells. I would suggest to rephrase that small section to make a clearer distinction between these two cell types, and/or add another sentence related solely to NK cells.
Author Response
We thank the Reviewer for the positive comment. We agree with all the suggestions and we modified the manuscript accordingly (Please see lines 125-127; 132-135 and Figure 1)
Reviewer 2 Report
The topic addressed by the authors of this manuscript is very interesting because it allows revealing the activities of the immune system affecting the gastrointestinal microbiota. However, conclusions should be drawn with great care and be mindful that they are influenced by the therapeutic interventions used (regarding human patients). Also, attention should be paid to the existing differences in the functioning of the immune system in animal models and humans. Especially since most laboratory animals have limited exposure to microorganisms and little diversity in their microbiota compared to wild ones. The recently introduced lab animal "co-housing" with pet-shop or wild captured ones experiments give the best view on the reserve that must be taken in drawing conclusions.
General comments:
The SFB colonization is age-dependent in humans, with the majority of individuals colonized within the first 2 years of life, but this colonization disappeared by the age of 3 years. This is very important and specific to SFB aspect of commensal colonization of human GIT. Since the bacteria play such a significant role in educating human immunity but latter disappear from the gut mucosa their involvement in development of immune system but not regulation throughout whole life should be clearly noted in the paper [you can reference in e.g. Yin Y, Wang Y, Zhu L, et al. Comparative analysis of the distribution of segmented filamentous bacteria in humans, mice and chickens. ISME J. 2013;7(3):615-621. doi:10.1038/ismej.2012.128].
In ch 3.2 (especially lines 141-144) the authors provide very simplified view on T-dependent and T-independent regulation of SIgA production in gut mucosal tissues. I understand this is introductory part to highlight important aspects in the area however, I would be valuable to mention that the subject was recently and thoroughly reviewed in Pietrzak, B.; Tomela, K.; Olejnik-Schmidt, A.; Mackiewicz, A.; Schmidt, M. Secretory IgA in Intestinal Mucosal Secretions as an Adaptive Barrier against Microbial Cells. Int. J. Mol. Sci. 2020, 21, 9254. https://doi.org/10.3390/ijms21239254.
In chapter 4.1 regarding SCID the authors should mention the recommendations and management of SCID patients, especially protective measures regarding feeding and housing which significantly reduce exposure to microorganisms therefore additionally impair colonization. As well as for other immune disorders mentioned if such recommendations (e.g. dietary) exist.
Specific comments:
Page 3, fig 1 (in both A and B panel) "B. Fragilis" should be "B. fragilis" (in italics),
Page 3, fig 1 (in B panel) secretory IgA should be abbreviated as "SIgA" according to current nomenclature
S-IgA and S-IgM are approved as abbreviations for secretory IgA and secretory IgM, respectively, to avoid confusion with surface immunoglobulins on B cells; accordingly, it is recommended that slg be used to refer to surface immunoglobulins. S also relates to "SC" as the abbreviation for secretory component, which is the distinctive feature of S-IgA (and S-IgM). A significant minority has expressed the view that the hyphenated forms (S-IgA and S-IgM) are unnecessary, and may choose to use unhyphenated forms (SIgA and SIgM).
Page 3, fig 1 description - in "(B) ... Bacteriodes Fragilis..." should be changed to "Bacteriodes fragilis" according to taxonomic nomenclature of species names the latter part should be in lower case (does not matter whether genus name is written in full or abbreviated).
full name of SFB on Page 3, fig 1 description "(B)" is in lower case and on page 3, lines 118-119 - "...Segmented Filamentous Bacteria..." is written in upper case. This needs to be unified throughout the text.
Page 4 line 140 consider using "commensal" instead of "symbiotic"
Page 5 line 196 "..., defined alpha diversity." consider changing to "defined by alpha diversity." or "as indicated by alpha diversity." for better readability of the sentence. And latter on I recommend defining the alpha diversity as a measure of diversity within community (in opposition to beta diversity as a measure of differences between communities) rather than providing the explanation how it is calculated.
Line 206 - "These mice presented with lower intestinal diversity..." make the sentence more clear rephrasing it and including that the diversity corresponds to intestinal microbiota composition as indicated by decreased alpha diversity measure. In present state it may look intriguing whether mice have any anatomical diversity regarding a part of intestine described as "lower" (as the differences in the anatomy of human and rodent GIT exist).
Line 244 - taxonomic (genus) names should italicized "...Lactobacillus and Bifidobacterium..."
Line 376 - "Actinobacteria" (phylum) name should be in italics
Line 377 - "spp." in regular font (not italics)
Line 377 - "Firmicutes" (phylum) name should be in italics
Line 378 - "Proteobacteria" (phylum) name should be in italics
Author Response
We thank the Reviewer for the positive comment. We agree with all the suggestions and we modified the manuscript accordingly.
In particular, please see lines:
- 146-147
- 172-174
- 189-194: as suggested, we added the concept of “wildling” mice
- 226-228
Moreover, we modified the figure and we corrected all the terms as suggested.